

# Using CALIOP to estimate cloud-field base height and its uncertainty: the Cloud Base Altitude Spatial Extrapolator (CBASE) algorithm and dataset

Johannes Mülmenstädt[1], Odran Sourdeval[1], David S. Henderson[2], Tristan S. L'Ecuyer[2], Claudia Unglaub[1], Leonore Jungandreas[1], Christoph Böhm[3], Lynn M. Russell[4], and Johannes Quaas[1]

[1]Institute of Meteorology, Universität Leipzig, Leipzig, Germany
[2]University of Wisconsin at Madison, Madison, Wisconsin, USA
[3]Institute for Geophysics and Meteorology, Universität zu Köln, Köln, Germany
[4]Scripps Institution of Oceanography, University of California, San Diego, San Diego, California, USA

**Correspondence:** Johannes Mülmenstädt (johannes.muelmenstaedt@uni-leipzig.de)

**Abstract.** A technique is presented that uses attenuated backscatter profiles from the CALIOP satellite lidar to estimate cloud base heights of lower-troposphere liquid clouds (cloud base height below approximately 3 km). Even when clouds are thick enough to attenuate the lidar beam (optical thickness $\tau \gtrsim 5$), the technique provides cloud base heights by treating the cloud base height of nearby thinner clouds as representative of the surrounding cloud field. Using ground-based ceilometer data, uncertainty estimates for the cloud base height product at retrieval resolution are derived as a function of various properties of the CALIOP lidar profiles. Evaluation of the predicted cloud base heights and their predicted uncertainty using a second, statistically independent, ceilometer dataset shows that cloud base heights and uncertainties are biased by less than 10%. Geographic distributions of cloud base height and its uncertainty are presented. In some regions, the uncertainty is found to be substantially smaller than the 480 m uncertainty assumed in the A-Train surface downwelling longwave estimate, potentially permitting the most uncertain of the radiative fluxes in the climate system to be better constrained. The cloud base dataset is available at https://doi.org/10.1594/WDCC/CBASE.

## 1 Introduction

The base height $z$ is an important geometric parameter of a cloud, controlling the cloud's longwave radiative emission, being required in the calculation of the cloud's subadiabaticity, and setting the level at which aerosol concentration and updraft speed determine the cloud's microphysical characteristics. However, due to the viewing geometry, it is also one of the most difficult cloud parameters to retrieve from satellite.

Multiple methods have been proposed for satellite determination of the cloud base height. Zhu et al. (2014) have used the Visible Infrared Imaging Radiometer Suite aboard the Suomi National Polar-orbiting Partnership satellite (VIIRS, Cao et al., 2014) to estimate cloud base temperature $T_b$ from the lowest cloud top temperature within a cloud cluster; a reanalysis temperature profile can be used to convert $T_b$ to $z$. Using an empirical relationship between geometric and optical thickness, Fitch et al. (2016) have obtained $z$ from VIIRS. Cloud geometric thickness (and therefore $z$ if the cloud top height is known)





can be inferred from increased spectral absorption by $O_2$ within cloud due to multiple scattering (Kokhanovsky and Rozanov, 2005; Lelli and Vountas, 2018). Stereoscopic determination of the height of the most reflective layer (Naud et al., 2005, 2007) in Multiangle Imaging Spectroradiometer data (MISR, Diner et al., 1998) yields information on $z$, as the lowest layer heights within a cloud cluster may correspond to the base of a cloud seen from its side. An evaluation of MISR techniques is in progress

(Böhm et al., 2018).

For analyses wishing to combine cloud base information with other cloud properties retrieved by A-Train satellites, these methods share the disadvantage that the required instruments are not part of the A-Train. Methods that are applicable to A-Train satellites are based on MODerate-resolution Imaging Spectroradiometer (MODIS, Platnick et al., 2017) cloud properties retrieved near cloud top and integrated along moist adiabats to determine the cloud thickness (Meerkoetter and Zinner, 2007)

or on active remote sensing by CloudSat (2B-GEOPROF, Marchand et al., 2008) or a combination of CloudSat and CALIOP (2B-GEOPROF-LIDAR, Mace and Zhang, 2014). Each of these has drawbacks. The MODIS-derived cloud thickness assumes adiabatic cloud profiles and therefore cannot be used to constrain subadiabaticity; the use of ancillary temperature profile estimates may also be problematic in many cases. CloudSat misses the small droplets at the base of nonprecipitating clouds, and retrievals are further degraded in the ground clutter region. CALIOP detects the bases of only the thinnest clouds ($\tau < 5$,

Mace and Zhang, 2014); frequently, it is desirable to know the base height of thick clouds as well.

In this paper, we revisit the CALIOP cloud base determination. This relies on one central assumption, namely that, because the lifting condensation level is approximately homogeneous within an airmass, the cloud bases retrieved by CALIOP for thin clouds may be a good proxy for the cloud base heights of an entire cloud field, including the optically thicker clouds within the field. We have designed an algorithm that extrapolates the CALIOP cloud base measurements into locations where CALIOP

attenuates before reaching cloud base. This algorithm is called Cloud Base Altitude Spatial Extrapolator (CBASE). In this paper we evaluate its performance by comparing CBASE $z$ against $z$ observed by ground-based ceilometers.

The cloud base of interest in this analysis is the base of the lowest cloud in each column. Even if CALIOP can also detect the base heights of other layers in multilayer situations, it is the base height of the lowest cloud that is of largest interest for many applications (e.g., surface radiation estimates).

Section 2 of this article describes the data sources used in determining and evaluating $z$. In Section 3 we describe the algorithm and evaluate its performance, including error statistics. The publicly available processed CBASE output is described in Section 4. We conclude in Section 5 with an outlook on the longstanding questions that the CBASE dataset can address.

## 2 Data

Two classes of data are used in this work: cloud lidar data, from which we intend to derive a global $z$ dataset; and ground-based

observations used as reference measurements of $z$ to train and evaluate the algorithm by which $z$ is determined from the satellite data.

Table 1 lists the URLs for all datasets used in this paper.



## 2.1 CALIOP VFM

The input satellite data to our analysis is from the Cloud–Aerosol Lidar with Orthogoncal Polarization (CALIOP, Winker et al., 2007) on board the Cloud–Aerosol Lidar and Infrared Pathfinder Satellite Observation (CALIPSO) satellite that is part of the A-Train satellite constellation (Stephens et al., 2002) on a sun-synchronous low-Earth orbit with equator crossings at

approximately 1330 hours local time. The cloud base product relies on the retrieved vertical feature mask (VFM, Vaughan et al., 2005). For each CALIOP lidar backscatter profile, the VFM identifies features such as clear air, cloud, aerosol, or planetary surface; this is termed the "feature type". (When the lidar beam is completely attenuated, this is reported as a feature type.) In addition to the feature type, the VFM records the degree of confidence in the identification ("none" to "high", termed the "feature type QA flag"); the thermodynamic phase of a layer identified as cloud as well as the degree of confidence therein

(termed "ice water phase" and "ice water phase QA flag"); the horizontal distance over which the algorithm had to average to identify a feature above noise and molecular atmospheric scattering ("horizontal averaging distance").

In the present analysis, we use VFM version 4.10 (CALIPSO Science Team, 2016), the current "standard" release, for the years 2007 and 2008. The VFM files are obtained from ICARE (http://www.icare.univ-lille1.fr/).

## 2.2 Airport ceilometers

For optimizing several parameters of the algorithm, for determining the expected cloud base uncertainty, and for evaluation of the trained algorithm, reference measurements of $z$ are required. The source of these "true" $z$ in this work is ground-based cloud observations at airports. Weather observations at airports are disseminated worldwide in aviation routine and special weather reports (METARs and SPECIs, collectively referred to as METARs henceforth, World Meteorological Organization, 2013). Apart from providing airport weather information for aviation, METAR data is used for assimilation into numerical

weather prediction (NWP) models (e.g., Benjamin et al., 2016; Dee et al., 2011). In many locations, $z$ reported in METARs is measured by a ceilometer over a period of time (tens of minutes) and then objectively grouped into cloud layers and their respective fractional coverages, using the temporal variation at a fixed point under an advected cloud field as a proxy for spatial variability of the cloud field (e.g., Heese et al., 2010). METAR data is widely distributed and archived; the data for the present analysis was downloaded from the Wunderground archive (https://www.wunderground.com/history/airport/).

In the United States, $z$ is mostly derived automatically by laser ceilometers that form part of Automated Surface Observing Stations (ASOS, National Oceanic and Atmospheric Administration, Department of Defense, Federal Aviation Administration, and United States Navy, 1998) system; see, e.g., An et al. (2017); Ikeda et al. (2017) for recent examples of ASOS application to deriving cloud climatologies or NWP model evaluation. In other parts of the world, the cloud bases may be estimated by human observers or may be omitted under certain conditions when the lowest cloud base is higher than 5000 feet, complicating

objective comparison to satellite $z$. To ensure that the ceilometer $z$ are of high and spatially uniform quality, we restrict ourselves to METARs from the contiguous continental United States.

There are 1645 stations throughout the continental USA that lie within 100 km of a CALIOP footprint. In normal operation, the time resolution of $z$ reports is 1 h, but during rapidly changing conditions, more frequent updates may be provided; for



comparison to satellite $z$, the ceilometer observation closest in time to the satellite overpass is used, provided that the time difference is less than 1 h. For training the algorithm, we use ceilometer observations from the year 2008. For unbiased evaluation of the algorithm performance, a statistically independent dataset is required; we use ceilometer observations from the same stations from the year 2007. Figure 1 shows the locations of these stations along with the number of satellite–

ceilometer $z$ coincidences and the closest co-location distance during the year 2007.

## 3  CBASE algorithm development and evaluation

The CBASE algorithm and evaluation proceed in four steps:

1. We determine $z$ from all CALIOP profiles where the surface generates a return, indicating that the lidar is not completely attenuated by cloud. We refer to this as the *local $z$* in the sense that it is local to the CALIOP profile.

2. Using ground-based ceilometer data, we determine quality of cloud base height depending on a number of properties of the CALIOP profile. Assuming those properties suffice to determine the quality of the $z$ estimate, we can then predict the quality of a cloud base as a function of those factors. The quality metric we use is the root mean square error (RMSE); the category RMSE determined from comparison to ceilometer $z$ then serves as the predicted $z$ uncertainty $\sigma$. In the language of machine learning, we refer to this step as *training* the algorithm on the ceilometer data to predict $z$ and $\sigma$.

3. Based on the predicted quality of each local cloud base, we either reject the local cloud base or combine it with other local cloud bases within a distance $D_{\mathrm{max}}$ of the point of interest (POI) to arrive at an estimate of $z$ and $\sigma$ at the POI.

4. Using a statistically independent validation dataset, we verify that the predicted $z$ and $\sigma$ are correct.

This section is divided into four subsections, one for each algorithm step enumerated above.

### 3.1  Determination of local $z$

Local $z$ is determined from the CALIOP VFM for each profile with a surface return. The rationale is that a surface return indicates that the lidar did not attenuate within the cloud, and that the lower limit of the layer identified as cloud therefore corresponds to the cloud base; Figure 2 illustrates the idea. For these profiles, the location, $z$, cloud top height, feature type between the cloud base and the surface, cloud thermodynamic phase, and associated quality assurance flags from the VFM algorithm are recorded.

### 3.2  Determination of local cloud base quality

We assess the quality of the CALIOP $z$ using the root mean square error (RMSE) with respect to the ceilometer-observed $\hat{z}$. The RMSE is defined as

$$E = \sqrt{\frac{1}{N} \sum_{i=1}^{N} (z_i - \hat{z})^2}. \tag{1}$$



The sum runs over all CALIOP profiles containing at least one cloud layer and a surface return that are within 100 km horizontal distance of the ceilometer, occurred within 3600 s of a ceilometer observation, and have their lowest CALIOP cloud feature within 3 km of the surface. Ceilometer observations are only used if the observation closest in time to the CALIPSO overpass contains a cloud within 3 km of the surface. This height limit is imposed because a subset of the ceilometers has a range limit

of 12500 feet, and all ceilometers report ceilings above 10000 feet with reduced granularity (500 feet); the 3 km threshold is safely below these ceilometer limitations and mimics the International Satellite Cloud Climatology Project (ISCCP, Rossow and Schiffer, 1999) definition of low cloud ($p > 680$ hPa).

The following metrics, which are useful for a qualitative assessment of the quality of the satellite cloud base, are also calculated but play no quantitative role in the algorithm:

**Correlation coefficient** between the CALIOP cloud base and ground-based observation of the cloud base. We use the Pearson correlation coefficient (ideally unity).

**Linear regression slope and intercept** (ideally 1 and 0, respectively).

**Retrieval bias,** defined as

$$\text{bias} = \frac{1}{N} \sum_{i=1}^{N} (z_i - \hat{z}),$$

(2)

(ideally 0)

CALIOP's ability to detect cloud base depends on the properties of the cloud. Therefore, we expect that the $z$ quality will vary between different cloud profiles. Measuring the quality as a function of various properties of the CALIOP column may allow us to predict the quality of other columns with the same combination of properties. The properties that are easily accessible in a single column and have substantial effects on quality are:

– horizontal distance $D$ from the ceilometer,

– number of column cloud bases within horizontal distance $D_{\max}$,

– CALIOP VFM feature quality assurance flag,

– geometric thickness of the lowest cloud layer,

– CALIOP thermodynamic phase determination of lowest cloud,

– feature type, if any, detected between the lowest cloud and the surface, and

– horizontal averaging distance required for CALIOP cloud feature detection.

For illustrative purposes, Figure 3 shows the joint distribution of CALIOP and ceilometer $z$ faceted by the CALIOP VFM feature quality assurance flag.

Based on determining the retrieval quality as a function of one variable at a time (integrating over the sample distribution of

the remaining variables), the following classes of CALIOP profiles are discarded:



- CALIOP VFM quality assurance worse than "high" ,

- "invalid" or "no signal" layers between the surface and the lowest cloud layer (indicating that although the surface may generate a detectable return, the lidar is sufficiently attenuated that the cloud base, which scatters less strongly than the surface, is unreliable),

- minimum CALIOP cloud detection horizontal averaging distance within the lowest cloud layer greater than 1 km (indicating that, although average cloud properties are known at the averaging length scale, those properties may not be representative of the particular CALIOP footprint under consideration), or

- thermodynamic phase of the lowest layer determined to be other than liquid by the CALIOP VFM algorithm (the reason for this is that not enough such columns exist to determine the RMSE reliably in each of the categories defined below).

The remaining variables are discretized roughly into quintiles of their distribution within the VFM dataset with the following boundaries:

- horizontal distance $D$ from the ceilometer, with boundaries 0, 40, 60, 75, 88, and 100 km (distance greater than 100 km is discarded),

- number of CALIOP columns $n$ with a cloud layer and a surface return within 100 km horizontal distance from the
15 ceilometer, with boundaries at 0, 175, 250, 325, 400 (multiplicities greater than 400 are accepted), and

- geometric thickness $\Delta z$ of the lowest cloud layer, with boundaries at 0, 0.25, 0.45, 0.625, and 1 km (thickness greater than 1 km is accepted).

We can now consider the joint distribution of CALIOP and ceilometer cloud bases for each combination of the above variables to derive the RMSE of each combination. For this comparison, we use $z$ above ground level (AGL); using $z$ above
20 mean sea level (MSL) would introduce an intrinsic correlation between satellite and ceilometer $z$ due to the varying terrain height, which would lead to an unrealistically positive assessment.

When calculating aggregate statistics such as the RMSE, a further consideration comes into play. $z$ above ground is positive-definite, which imposes a physical phase-space boundary. Due to this boundary, the satellite $z$ estimate is intrinsically biased high (negative excursions may be removed by the phase-space boundary, but positive excursions are not), and the bias decreases
with increasing satellite $z$ estimate (when true $z$ is high, it is less likely that measurement error would lead to a negative AGL $z$). Since this effect constitutes a bias rather than a random error, it cannot be eliminated by averaging over large sample sizes, but instead needs to be corrected for. Since the effect is nonlinear in $z$, a nonlinear correction method is required; our choice of nonlinear bias correction is an $\varepsilon$-regression support vector machine (SVM, Chang and Lin, 2011).

Following bias correction, the sample RMSE is calculated for each combination of $D$, $n$, and $\Delta z$. The sample RMSE is taken
as an estimate of the statistical uncertainty $\sigma(D, n, \Delta z)$ on the CALIOP $z$.



### 3.3 Combination of local cloud bases

CALIOP $z$ only exists sporadically, when CALIOP happens to hit a sufficiently thin cloud. To infer the $z$ at a point of interest (POI) that does not necessarily coincide with the location of a thin-cloud CALIOP profile, we proceed as follows. We first select all local CALIOP $z$ measurements within a horizontal distance $D_{max} = 100$ km of the POI that satisfy the additional quality cuts described in Section 3.2.

For each remaining local $z_i$, we determine the predicted uncertainty $\sigma_i$ based on the categories established in the previous section. We determine a combined $z$

$$z = \frac{\sum\limits_{i}^{n} w_i z_i}{\sum\limits_{i}^{n} w_i} \tag{3}$$

with weights

$$w_i = \frac{1}{\sigma_i^2} \tag{4}$$

(optimal weights for uncorrelated least-squares). In practice, the individual measurements of cloud base are highly correlated with fairly similar $\sigma_i$. The cloud base estimate by Eq. (3) with weights given by Eq. (4) remains unbiased even in the presence of correlations. However, for the combined cloud base uncertainty, the uncorrelated weights would yield a biased estimate in the presence of correlations. The expression

$$\sigma^2 = \frac{1}{n} \sum\limits_{i}^{n} \sigma_i^2 \tag{5}$$

yields acceptable results, as would be expected for highly correlated and fairly similar $\sigma_i$.

### 3.4 Evaluation of CBASE $z$ and $\sigma$

Having trained the algorithm on data from the year 2008, we evaluate it using a statistically independent dataset from the year 2007. In the evaluation dataset, the "true" (i.e., ceilometer-measured) $\hat{z}$ is known in addition to the estimated $z$ and the estimated cloud base uncertainty $\sigma$, determined according to the procedure described in the previous section. Figure 4 shows the joint distribution of CBASE and ceilometer-observed $z$.

For satellite-derived measurements of $z$ that are unbiased with respect to the ceilometer-observed $\hat{z}$ and have correctly estimated uncertainties $\sigma$, the pdf of the quantity $(z - \hat{z})/\sigma$ has zero mean and unit standard deviation. In our evaluation dataset, we find a mean of 0.04 and a standard deviation of 1.06, shown in Figure 5; this corresponds to a $z$ bias of 4% and uncertainty bias of 6%, both relative to the predicted uncertainty. Thus, we find that both the cloud base estimate and the uncertainty estimate are unbiased at better than the 10% level.

As a further test of the reliability of the expected uncertainty, we divide the validation dataset into deciles of the expected uncertainty. Table 3 shows that the actual RMSE within each decile is within 10% of the expected uncertainty (with the exception of the highest-uncertainty decile) and that linear regressions within each decile are close to the one-to-one line.



It is possible that $z$ estimates outside North America could have greater biases or greater uncertainty than this evaluation leads us to believe. This would be the case if continental clouds over North America are not representative of clouds elsewhere in a way that is not accounted for by the cloud properties considered by the uncertainty estimate. Since the validation sample spans an entire year on a continental scale, we expect that most cloud morphologies are included. However, cloud types that occur predominantly over ocean, namely marine stratocumulus with horizontally extensive but vertically thin liquid-phase anvils, present a particular challenge to the method. Due to the typical $z$ uncertainty of several hundred m, the method is unlikely to be applied to stratocumulus cloud; nevertheless, a marine-cloud validation dataset would be desirable. For the present work, no suitable marine-cloud evaluation dataset was available; ship-based $z$ observations were either based on human observers with coarse vertical resolution and a precision that is difficult to characterize; or available only over a limited duration at limited locations, resulting in a severely statistics-limited set of coincidences with the CALIOP track.

## 4   Results and data product availability

Geographic distributions of the mean $z$ are shown for daytime and nighttime Calipso overpasses in Figure 6. Over most of the globe, especially over land, daytime $z$ is higher than nighttime $z$, consistent with the diurnal deepening of the planetary boundary layer. Figures 7 and 8 show the distribution of $z$ uncertainties. A larger fraction of nighttime cloud bases falls into the lowest uncertainty range (200 to 350 m), while the the nighttime uncertainty distribution peaks slightly higher than the daytime uncertainty distribution and features a substantial tail above 500 m that is not present in the daytime distribution. CALIOP benefits from higher signal to noise ratio during nighttime, which may lead to lower $\sigma$, but this effect would be convoluted with potential differences between daytime and nighttime clouds that can lead to different $z$ uncertainties.

Comparison with 2B-GEOPROF-LIDAR cloud bases is shown in Figure 9. 2B-GEOPROF-LIDAR distinguishes between radar-only, lidar-only, and radar–lidar combined cloud bases; the latter category is rare for warm clouds and is not shown. For radar-only clouds, the mean error is large because the radar $z$ predominantly clusters around the top of the ground clutter region with little dependence on the actual $z$. Lidar-only 2B-GEOPROF-LIDAR cloud base performs comparably to the CBASE cloud base on average; this is to be expected, as the underlying physical measurement is the same. Unlike 2B-GEOPROF-LIDAR, CBASE provides a validated uncertainty estimate, which allows an analysis to select only low-uncertainty cases or to statistically weight $z$ according to uncertainty, as appropriate for the application.

As an example application, we consider the surface downwelling longwave radiation $F^{\downarrow}_{\mathrm{surf}}$, which is strongly affected by cloud base temperature. Henderson et al. (2013) derive a global $F^{\downarrow}_{\mathrm{surf}}$ sensitivity to $z$ of 1.5 W m$^{-2}$ for a $z$ perturbation of one CloudSat height bin (240 m); as Table 4 and Figure 9 show, the CloudSat $\sigma$ specifically for the low clouds at the focus of the present work is likely greater than 240 m, which corroborates the 480 m uncertainty estimate of Kato et al. (2011). To arrive at a conservative estimate of the improvement in $F^{\downarrow}_{\mathrm{surf}}$ uncertainty that might be possible by utilizing the CBASE predicted $\sigma$, we compare two $F^{\downarrow}_{\mathrm{surf}}$ uncertainty distributions: one based on a globally constant 400 m $\sigma$ (Figure 10a) and one with the CBASE $\sigma$ achievable by selecting the highest-quality percentile of the CBASE dataset (Figure 10b). This selection provides a $\sigma$ of approximately 250 m in the extratropics as well as the nighttime tropical continents and SCu regions, and approximately





400 m throughout the tropics during daytime, according to Figure 8. Globally, the $F^{\downarrow}_{\text{surf}}$ uncertainty is reduced from 3.1 W m$^{-2}$ to 1.8 W m$^{-2}$, assuming that the $z$ uncertainty contribution to the $F^{\downarrow}_{\text{surf}}$ uncertainty is dominated by low clouds. Improvements are especially large in the marine stratocumulus regions and the extratropical oceans, where extensive low cloud often overlies cool air with relatively low longwave emission by water vapor. The selection reduces the available statistics by a factor of 100,

but analyses based on A-Train data are usually not statistics-limited.

The CBASE $z$ and $\sigma$ dataset (Mülmenstädt et al., 2018) spanning the years 2007 and 2008 is freely available at Deutsches Klimarechenzentrum (DKRZ) under the DOI https://doi.org/10.1594/WDCC/CBASE. The dataset is provided in two spatial resolutions corresponding to different window sizes within which CALIOP profiles are combined: $D_{\text{max}} = 40$ km and $D_{\text{max}} = 100$ km. CBASE provides two files for each CALIOP VFM input file: one using a 40 km window to detect the cloud field base

height, and one using a 100 km window. (The input CALIOP VFM dataset is organized by the daytime (D)/nighttime (N) half of each orbit.) The file name pattern is `CBASE-{40|100}.<date>T<time>{D|N}.nc` (identical to the input CALIOP VFM file name with the exception of the product name and file-type extension). Files are organized into subdirectories by half-orbit start date. In case no cloud base heights are detected within a half-orbit, no output file is produced. Otherwise, each CALIOP VFM input file results in a 40 km-resolution and a 100-km resolution CBASE file. The measurement quality is

reported as a quantitative uncertainty estimate for each cloud field.

Following publication of the final paper, the source code used to produce the dataset and evaluation plots will be made public at https://github.com/jmuelmen.

## 5    Conclusions

We have presented the CBASE algorithm, which derives the cloud base height $z$ from CALIOP lidar profiles. This algorithm

produces $z$ not only for thin clouds but also for clouds thick enough to attenuate the lidar (optical thickness $\tau \gtrsim 5$), based on the assumed mesoscale homogeneity of cloud base height within an airmass. In addition to the $z$ estimate, the CBASE algorithm supplies an expected uncertainty $\sigma$ on the $z$. The CBASE dataset is available for the years 2007 and 2008 at https://doi.org/10.1594/WDCC/CBASE.

CBASE $z$ and $\sigma$ have been evaluated using ground-based airport ceilometers over the contiguous United States using a data

sample unbiased by the training of the algorithm. The evaluation showed that $z$ and $\sigma$ are unbiased at the better than 10% level: the bias on the $z$ is 4%, and the bias on the uncertainty is 6%, both relative to the expected uncertainty.

The performance of CBASE $z$ is similar to that of 2B-GEOPROF-LIDAR lidar-only $z$, which is based on the same underlying physical measurement. However, the validated $z$ uncertainty provided by CBASE allows for selection of only accurate cloud base heights or for statistically weighting of $z$ according to expected uncertainty. This, in turn, makes the CBASE $z$ useful

for pressing problems in climate research that require accurate knowledge of cloud geometry, such as surface downwelling longwave radiation or cloud subadiabaticity, which will be presented in future work.



*Acknowledgements.* We thank Patric Seifert and Albert Ansmann for valuable suggestions on the algorithm; ICARE for hosting the CALIOP VFM dataset, which was originally obtained from the NASA Langley Research Center Atmospheric Science Data Center; and DKRZ for computing and data hosting. This research was funded by the European Union under ERC Starting Grant QUAERERE, grant agreement 306284, and by the United States National Science Foundation under grant agreements AGS-1013423 and AGS-1048995.



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



**Table 1.** Data sources used in this analysis

| Data product | URL |
| --- | --- |
| CALIOP VFM | http://www.icare.univ-lille1.fr/archive?dir=CALIOP/VFM.v4.10/ |
| ASOS locations | http://www.rap.ucar.edu/weather/surface/stations.txt |
| METAR data | https://www.wunderground.com/history/airport/[1] |
| CBASE | https://doi.org/10.1594/WDCC/CBASE |

[1] As a first step, ASOS station identifiers within 100 km great-circle distance of a CALIOP footprint are identified; as a second step, the ICAO identifier of the ASOS station is then used to query the Wunderground METAR database.

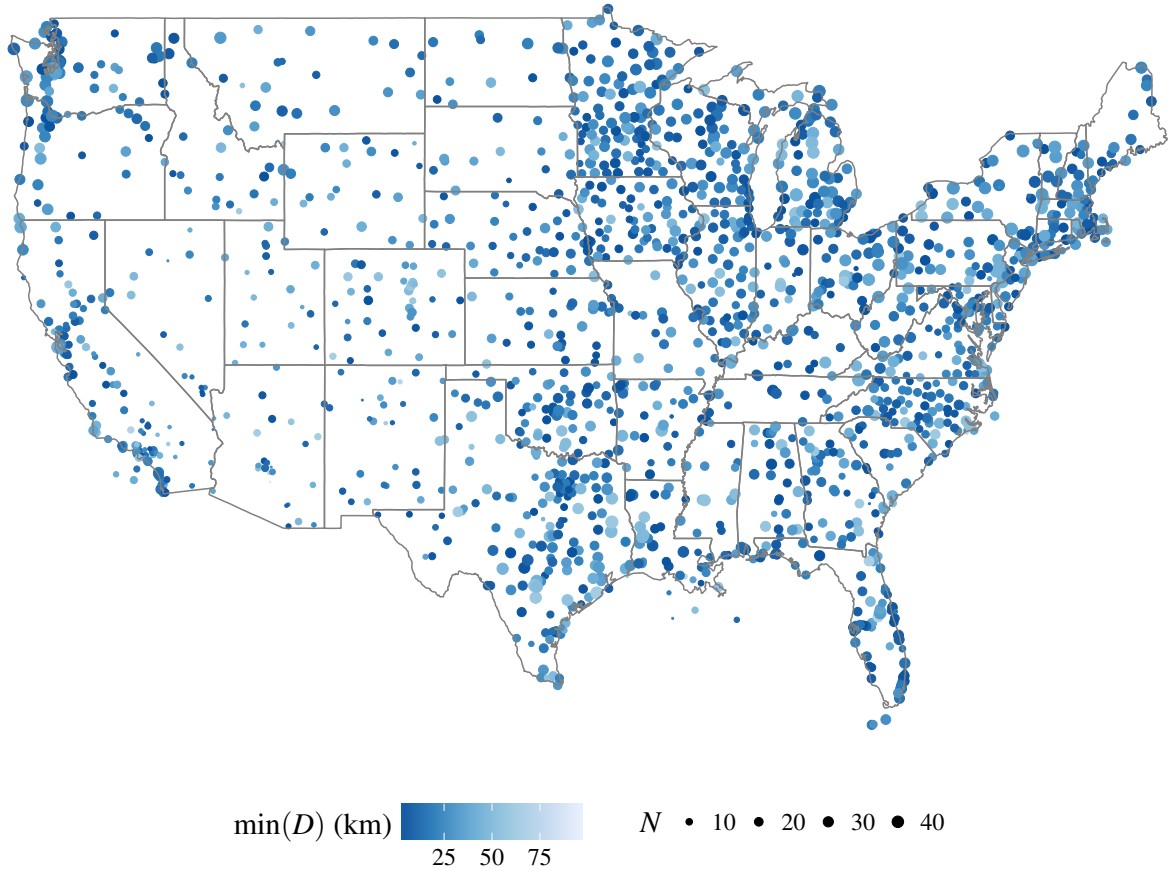

**Figure 1.** ASOS ceilometers used for CBASE $z$ evaluation. The size of the marker indicates the number of satellite–ceilometer $z$ coincidences during the year 2007. Color indicates the closest co-location distance achieved in 2007.



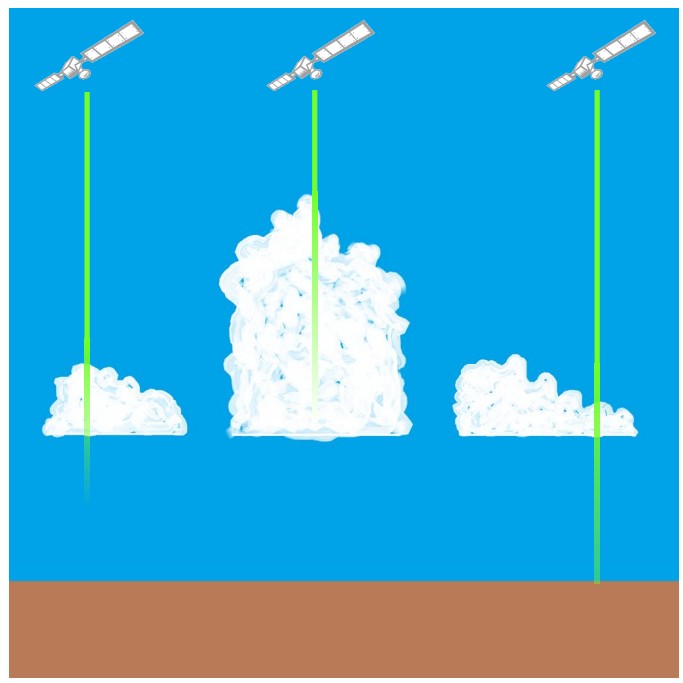

**Figure 2.** Schematic of CALIOP cloud base determination and evaluation strategy. In optically thick clouds (left and center), the lidar attenuates significantly within the cloud, rendering the cloud base information unreliable. However, $z$ of thin clouds (right) can be used as a proxy for thick clouds in a cloud field with homogeneous $z$.

**Table 2.** Statistics of the relationship between ceilometer and CALIOP cloud base height faceted by CALIOP VFM QA flag. Shown are the number of CALIOP profiles $n$, the product-moment correlation coefficient $r$ between CALIOP and ceilometer $z$, the RMSE, bias, and linear least-squares fit parameters.

| QA flag | $n$ | $r$ | RMSE (m) | bias (m) | fit |
|---|---|---|---|---|---|
| none | 1410553 | 0.192 | $1.05 \times 10^3$ | −471. | $\hat{z} = 0.193z + 1.03 \times 10^3$ m |
| low | 301250 | 0.471 | 710. | −115. | $\hat{z} = 0.456z + 650.$ m |
| medium | 212723 | 0.502 | 707. | −77.1 | $\hat{z} = 0.476z + 602.$ m |
| high | 2877967 | 0.554 | 629. | 9.85 | $\hat{z} = 0.526z + 485.$ m |



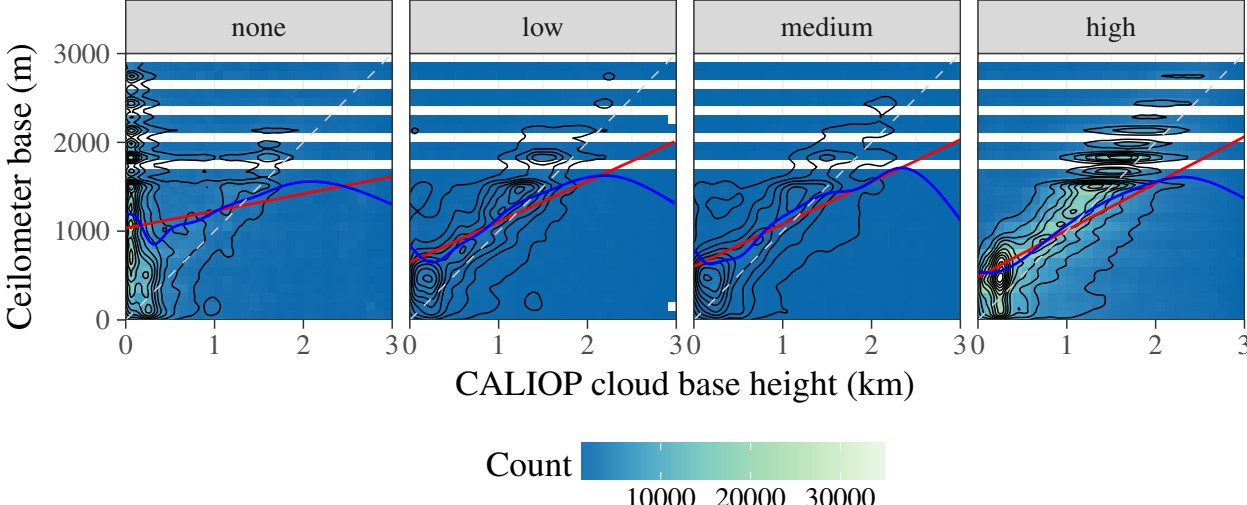

**Figure 3.** Scatter plots of CALIOP versus ceilometer local cloud base height faceted by the CALIOP VFM QA flag. Color indicates the number of CALIOP profiles within each bin of ceilometer and CALIOP $z$; black lines are contours of the empirical joint probability density; the red line is a linear least-squares fit, with 95% confidence interval shaded in light red; the blue line is a generalized additive model regression (Wood, 2011), with 95% confidence interval shaded in light blue; the dashed gray line is the one-to-one line. Statistics of the relationship between CALIOP and ceilometer base heights are provided in Table 2.

**Table 3.** CBASE cloud base statistics by decile of predicted uncertainty; see Table 2 for a description of the statistics provided.

| Predicted $\sigma$ (m) | $n$ | $r$ | RMSE (m) | bias (m) | fit |
|---|---|---|---|---|---|
| (167,427] | 2624 | 0.741 | 404. | −46.9 | $\hat{z} = 1.03z + 28.0$ m |
| (427,453] | 2624 | 0.719 | 429. | −28.4 | $\hat{z} = 1.06z − 32.0$ m |
| (453,469] | 2624 | 0.703 | 461. | −18.8 | $\hat{z} = 1.09z − 87.7$ m |
| (469,484] | 2624 | 0.685 | 463. | −17.8 | $\hat{z} = 1.03z − 18.3$ m |
| (484,497] | 2624 | 0.628 | 506. | −6.06 | $\hat{z} = 0.976z + 33.4$ m |
| (497,508] | 2624 | 0.574 | 547. | −8.73 | $\hat{z} = 0.986z + 25.5$ m |
| (508,522] | 2624 | 0.596 | 547. | −14.1 | $\hat{z} = 1.01z + 5.37$ m |
| (522,541] | 2624 | 0.572 | 562. | −9.26 | $\hat{z} = 0.967z + 49.6$ m |
| (541,573] | 2624 | 0.502 | 639. | −22.7 | $\hat{z} = 0.939z + 96.8$ m |
| (573,748] | 2624 | 0.447 | 720. | 7.36 | $\hat{z} = 0.829z + 197.$ m |



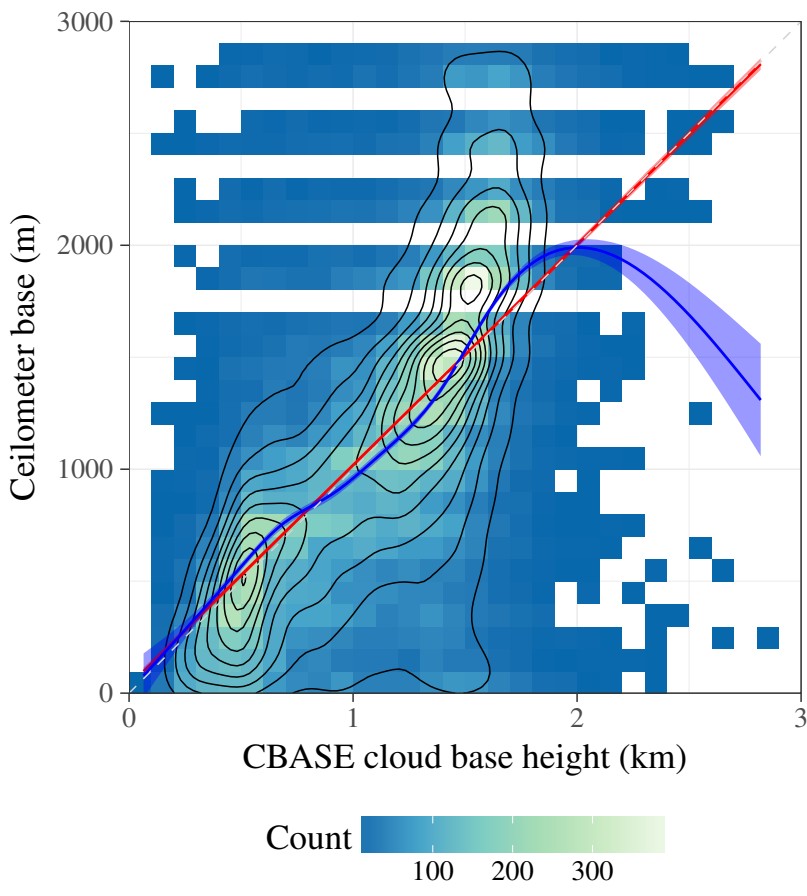

**Figure 4.** Scatter plot of CBASE versus ceilometer $z$ for all A-Train overpasses over the CONUS available for 2007; for description of the plot elements, see Figure 3. The linear fit has slope 0.98 and intercept 33.96 m.

**Table 4.** Statistics of the relationship between ceilometer and 2B-GEOPROF-LIDAR $z$; see Table 2 for a description of the statistics provided.

| Base type | $n$ | $r$ | RMSE (m) | bias (m) | fit |
|---|---|---|---|---|---|
| Radar | 15061 | 0.265 | 782. | 98.1 | $\hat{z} = 0.461z + 466.$ m |
| Lidar | 12813 | 0.564 | 594. | 16.3 | $\hat{z} = 0.555z + 399.$ m |





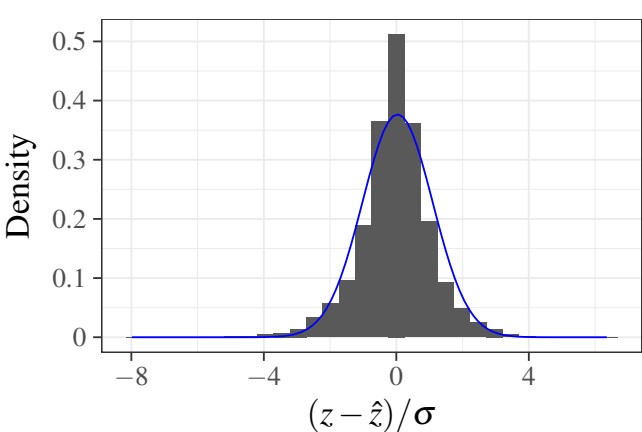

**Figure 5.** Distribution function of cloud base error divided by predicted uncertainty; for the ideal case of unbiased $z$ and unbiased uncertainty, the distribution would be Gaussian with zero mean and unit standard deviation. The superimposed least-squares Gaussian fit (blue line) has mean 0.04 and standard deviation 1.06.





**Figure 6.** Geographic distribution of mean $z$ above ground level. Statistics are calculated within each $5° \times 5°$ latitude–longitude box, and separately for CALIOP daytime (top) and nighttime (bottom) overpasses.





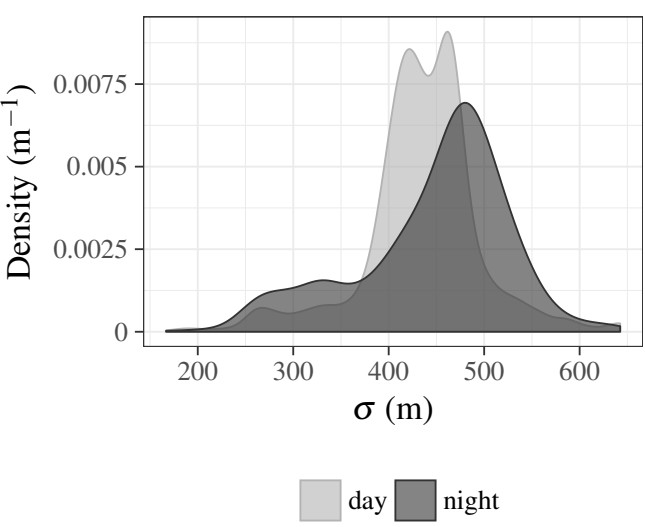

**Figure 7.** Distribution of predicted $z$ uncertainty $\sigma$.





**Figure 8.** Cloud base uncertainty quantiles. Statistics are calculated within each $5° \times 5°$ latitude–longitude box. The left (right) column shows statistics of daytime (nighttime) retrievals; daytime and nighttime are defined by the CALIOP VFM product.

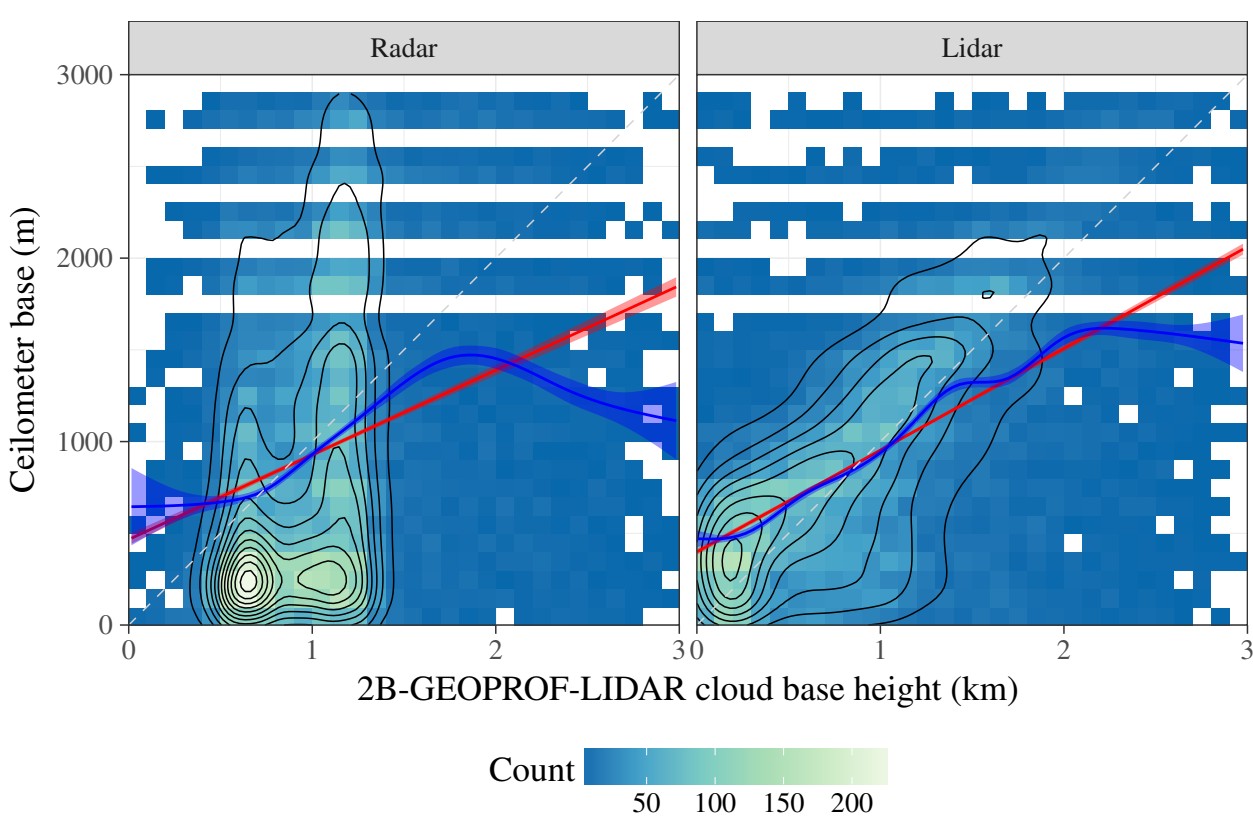

**Figure 9.** Scatter plot of 2B-GEOPROF-LIDAR versus ceilometer *z* faceted by the source of the cloud base (radar-only or lidar-only; due to their rare occurrence, combined radar–lidar base heights are not shown). For description of the plot elements, see Figure 3. Statistics of the relationship between 2B-GEOPROF-LIDAR and ceilometer base heights are provided in Table 4.



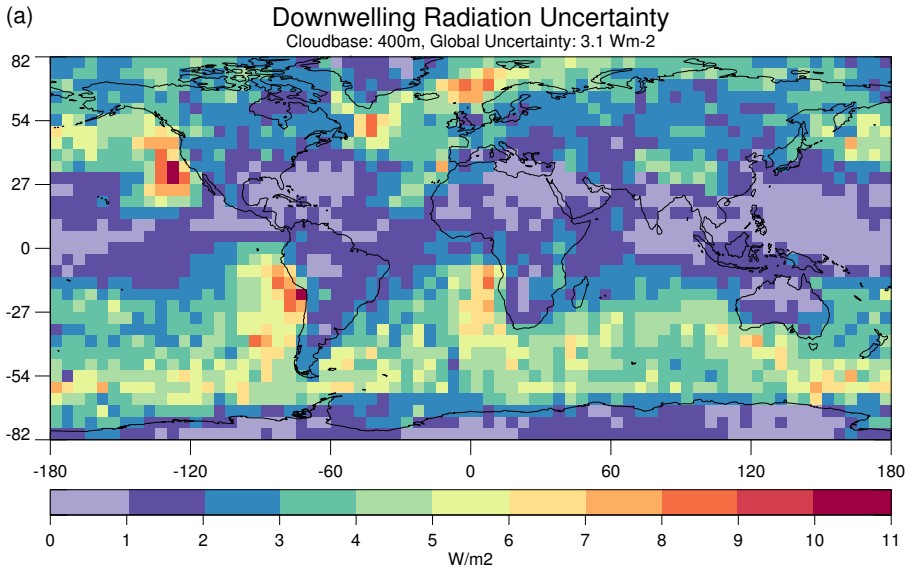

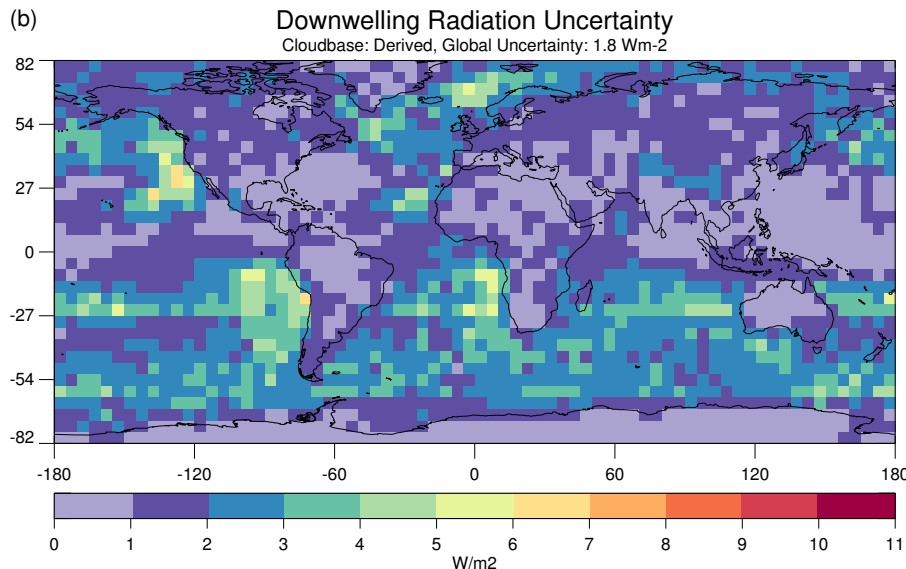

**Figure 10.** Uncertainty on the surface downwelling longwave radiation $F_{surf}^{\downarrow}$ under two assumptions on $z$ uncertainty: (a) constant 400 m uncertainty globally and (b) uncertainty achievable by selecting a high-quality subset of CBASE $z$.