# Peer review of "Using CALIOP to estimate cloud-field base height and its uncertainty: the Cloud Base Altitude Spatial Extrapolator (CBASE) algorithm and dataset"

_Earth System Science Data, 2018_

## Referee Comment (RC1) · Anonymous Referee #1 · 22 May 2018

A new technique is presented to retrieve water clouds base heights in the lower troposphere (base below about 3 km). The technique uses CALIOP base heights reported in the V4.10 VFM product and base heights from ground-based laser ceilometers that are part of the Automated Surface Observing Stations in the United States of America. The algorithm is trained using collocated (< 100 km and < 1h) water clouds base heights from the CALIOP satellite and from the reference ground-based ceilometers in the year 2008 under selected conditions. The algorithm supplies estimated uncertainties shown to be in some regions smaller than currently assumed uncertainties,

thereby offering the possibility to reduce the uncertainty in the computation of surface downwelling longwave radiation. Results as well as the data products are presented. The manuscript is well organized and well written, and the data product is of interest for the scientific community. This manuscript is suitable for publication in this journal. However, I have several comments that should be addressed before the manuscript can be published.

**1: Figure 3: I suggest to give explanations and to add discussions: a) What is CALIOP cloud base height in the X-axis? Is it the mean value of the N relevant local CALIOP $z_i$? Please clarify. b) Please specify whether height is above sea level or above ground level. c) I don't see the 95% confidence intervals shaded in light red and in light blue. d) Looking at the contours of the joint probability density for instance in the right hand-side plot (high), it looks like the agreement between ceilometer and CALIOP cloud base heights for bases larger than 1.5 km is better than indicated by the red and blue lines. Can you comment? e) It looks like Fig. 3 (high) has been obtained before discarding the classes of CALIOP profiles listed page 6, lines 1 to 9. If this is correct, it would be very informative to show scatter plots as in Fig. 3 (high), but after discarding these profiles.**

**2: Page 4, Eq. (1): why are the authors using a new notation "E" for RMSE?**

**3: Page 6, lines 22-28: a) Please explain how CALIOP z is converted to "z Above Ground" (which reference for the elevation maps?). b) For more clarity, it would be useful to use different notations for "Above Sea Level z" and "Above Ground Level z", for both the ceilometer and CALIOP. c) I am not sure why the satellite z estimate is intrinsically biased high "due to this boundary". Do you mean that the technique requires the local CALIOP "Above Ground Level" $z_i$ to be positive? d) I could not figure out how these biases are corrected (lines 26-28). Please develop and quantify. These bias corrections seem to be an important part of the algorithm training.**

**4: Page 6, lines 29-30: a) Do you confirm that you are introducing a new notation for**

RMSE, which is now "sigma"? b) Can you elaborate? For instance: what is the range of values for sigma(D,n,Dz)? What is the sensitivity of sigma to D, n, and Dz?

**5: Page 7, Equations 3 and 4: a) If I understand correctly, "z" in Eq. (3) is CBASE_z. Please clarify and use a specific notation for the different quantities. Indeed, "z" is used several times throughout the manuscript, but with different meanings. b) Please define "n" in Eq. (3). c) In Eq. (4), is sigma-i actually sigma-i(Di,ni,Dzi), with Di, ni, and Dzi associated to CALIOP local zi? How is ni defined for a local zi? d) Can you discuss the impact of the training? For instance: how do CBASE_z and CALIOP mean base height compare for the year 2008 used to train the algorithm? How did you train the algorithm to have mean (CBASE_z -ceilometer zĚȨ) equal to zero (as suggested by statement line 23, page 7)?**

**6: Page 8, lines 12-18: My understanding is that the algorithm training and the verification presented in Sect. 3.4 using the 2007 data set have been carried out with no distinction between nighttime and daytime data. Did you investigate whether sigma-i (Di,ni,Dzi) are the same for nighttime and daytime data? I wonder whether the differences between the nighttime and daytime CBASE_z highlighted here could be due in part to the fact that the algorithm training combines daytime and nighttime data.**

**7: Page 8, lines 19-25: a) Figure 9 seems out of place. In my opinion, this discussion could be earlier in the manuscript. However, comparisons of 2B-GEOPROF-LIDAR and CBASE base altitudes would be informative. b) Please describe the "underlying physical measurement" in 2B-GEOPROF-LIDAR that explain the similarity of lidar-only 2B-GEOPROF-LIDAR and CBASE cloud bases. c) Are you implying that for lidar-only cases, cloud bases reported in 2B-GEOPROF-LIDAR differ from those reported in the V4.10 CALIOP VFM and are in better agreement with CBASE? Does 2B-GEOPROF-LIDAR use V4.10 CALIOP data?**

**8: Page 9, lines 20-21: did the authors investigate how CBASE_z and CALIOP base altitude compare for clouds that are thick enough to attenuate the lidar laser beam?**

**9 Page 9, line 27: was stated but not shown.**

---

## Referee Comment (RC2) · Anonymous Referee #2 · 10 Jul 2018

In summary, I do not recommend this paper for publication in its current state. The methodology is questionable. The conclusions are highly qualified, and its not clear to me what exactly the authors have contributed to the science. In fact, its not clear to me at all why this methodology is even needed. Let me summarize my primary scientific concerns:

- Its not clear why you'd need to resolve cloud base from CALIOP when CloudSat can do it. That's the whole point of the synergy between the lidar/radar. Further, its not clear to me why, if you can resolve cloud base with CALIOP for an optically-tenuous clouds,

why you'd need an algorithm to understand potential correlation and uncertainty, and thus who'd even use it? If this paper is going to be publishable, the reviewers needs to go back and very considerately make the case that answers these questions. In my opinion, they have not done so beyond a threshold necessary for publication.

- Training of your dataset relative to ground-based ceilometers, as you even state, limits your application to a very small set of cloud types. The authors seem aware of this, but only speculate as to its impact. I ask again, who is the customer for this dataset, and how will it advance any scientific interest? How was the 100 km threshold for collocating with ceilometers chosen? What is the correlation length of cloud base spatially so as to justify such a choice? What is the impact on your results if you vary that threshold? There have been efforts (Omar et al. 2013 for aerosols...JGR-A) to collocate CALIOP with ground-based sun photmeters. They came up with something like 1500 suitable collocations under a much more stringent set of temporal and spatial thresholds. What you're trying to do requires far more justification and scientific basis, as it goes against conventional/proven thinking otherwise.

I recognize that as this is a Discussions page, that the likelihood is that the authors will be afforded opportunity to respond. That's fine. I caution, however, that if this were a more standard journal, I would be recommending an outright rejection.

---

## Author Comment (AC1) · 9 Aug 2018

*The manuscript is well organized and well written, and the data product is of interest for the scientific community. This manuscript is suitable for publication in this journal. However, I have several comments that should be addressed before the manuscript can be published.*

We thank the reviewer for his or her insightful reading of the manuscript and useful suggestions for improvements.

*#1: Figure 3: I suggest to give explanations and to add discussions: a) What is CALIOP cloud base height in the X-axis? Is it the mean value of the N relevant local CALIOP zi? Please clarify.*

There is an entry for each local CALIOP cloud base estimate within the collocation criteria (hence the large values of $n$ in the table). We have included this missing information in the caption.

*b) Please specify whether height is above sea level or above ground level.*

All heights are above ground (to avoid spurious correlation between satellite and ceilometer cloud base due to terrain height). We have rephrased the first mention of this in the manuscript from "For this comparison, we use $z$ above ground level (AGL); ..." to "Throughout this work, we use $z$ above ground level (AGL); ..." to make it clearer that this statement applies to the entire manuscript. We have also included "AGL" in the axis labels in all figures.

*c) I don't see the 95% confidence intervals shaded in light red and in light blue.*

Thanks for pointing this out. Due to the essentially infinite A-Train statistics, the width of the confidence intervals is smaller than the line width. Since this type of plot appears several times and we refer to the first instance's caption for a full description, we would like to keep the full description here. We have added a note that the confidence intervals are narrower than the line width in this figure.

*d) Looking at the contours of the joint probability density for instance in the right hand-side plot (high), it looks like the agreement between ceilometer and CALIOP cloud base heights for bases larger than 1.5 km is better than indicated by the red and blue lines. Can you comment?*

The straight-line fit that tries to accommodate high and low cloud bases at the same time is problematic when there is curvature in the relationship between satellite and ceilometer cloud base. This is why we also include the GAM regression (blue line),

whose slope and intercept need not be constant. The GAM regression is still below the 1-to-1 line at $z_c \approx 1.5$ km because there is a nonnegligible population of clouds with low ceilometer base and high CALIOP base (see the bulge in the outermost density contour near 0 ceilometer base height and 1.5 km CALIOP base height). These cases mostly occur when the ceilometer reports multiple layers of fractional cloudiness, where the satellite perspective may cause the algorithm to select one of the higher layers as the "base" because they obscure the lowermost layer. We are investigating whether the number of layers reported by 2B-GEOPROF-LIDAR can be used as another predictor variable for cloud base uncertainty as a future improvement to CBASE.

*e) It looks like Fig. 3 (high) has been obtained before discarding the classes of CALIOP profiles listed page 6, lines 1 to 9. If this is correct, it would be very informative to show scatter plots as in Fig. 3 (high), but after discarding these profiles.*

That is correct. At the point in the text where Figure 3 is discussed, the additional requirements have not yet been introduced. We agree with the reviewer that it is interesting to see the improvement in the scatter as the selection criteria on CALIOP columns are tightened. We have included this plot as a new Figure 4, discussed at the point in the manuscript when the selection criteria are introduced.

*#2: Page 4, Eq. (1): why are the authors using a new notation "E" for RMSE?*

We defined $E$ out of a general aversion to multiletter variable names. However, since we do not use $E$ again (and use "RMSE" as column heading in the tables), we agree with the reviewer that it would be preferable not to introduce new notation. We have changed $E$ to RMSE in the equation.

*#3: Page 6, lines 22-28: a) Please explain how CALIOP z is converted to "z Above Ground" (which reference for the elevation maps?).*

We subtract the elevation provided in the CALIOP VFM data files from the MSL cloud height. As of V4.10 (the version used here), the CALIOP L2

products use the "CloudSat DEM", according to https://eosweb.larc.nasa.gov/news/
caliop-v410-l1-l2-release-announcement, which contains reduced artifacts compared
to GTOPO30. This DEM is also mentioned in the CloudSat R05 1A-AUX description
(ftp://ftp.cira.colostate.edu/ftp/Partain/1A-AUX_PDICD_5.0.doc), but we could not find
a reference for it. We note that DEM errors may spuriously increase the apparent
disagreement between satellite and ceilometer cloud base, but this effect should be
very small to the spurious correlation enhancement that would occur if we used MSL
heights. We have added the following passage to the manuscript: "To convert cloud
base heights to AGL height, we subtract the surface elevation contained in the CALIOP
VFM data files, which in turn comes from the CloudSat R05 surface digital elevation
model."

*b) For more clarity, it would be useful to use different notations for "Above Sea Level z"
and "Above Ground Level z", for both the ceilometer and CALIOP.*

All heights are above ground (see our response above), so the "for this comparison"
qualifier is misleading. We have changed it to "Throughout this manuscript".

*c) I am not sure why the satellite z estimate is intrinsically biased high "due to this
boundary". Do you mean that the technique requires the local CALIOP "Above Ground
Level" zi to be positive?*

Not the technique, but rather the physics, because there are no clouds below the sur-
face. Consider the idealized case where the satellite provides an unbiased estimate of
cloud base height with random error $\epsilon$ symmetric about 0. When $z \gg \epsilon$, the average of
many satellite profiles will converge to the true cloud base height. However, when $z$ is
comparable to $\epsilon$, the sample will preferentially contain cases where the random error
is positive, because false detection of cloud below the surface is physically impossible.
In that case, the average of many satellite profiles will not converge to the true cloud
base height unless a correction is applied.

*d) I could not figure out how these biases are corrected (lines 26-28). Please develop*

*and quantify. These bias corrections seem to be an important part of the algorithm training.*

We agree that the correction needs to be explained in greater detail. We have added a brief description of support vector machines in general and attempted to explain the correction method by analogy to a linear correction.

Quantification is nontrivial because the correction is a function of $z_c$, $D$, $n$, and $\Delta z$ (and $D$, $n$, and $\Delta z$ can be correlated). To reduce the dimensionality of this multivariate correction, we have used the training dataset (with its joint distribution of $z_c$, $D$, $n$, and $\Delta z$) to calculate an ensemble of correction factors that can be expected in a realistic sample of clouds. We have added discussion and a plot of these correction factors to the manuscript.

The correction is desirable because otherwise (a) the average over many CBASE estimates would not converge to the true cloud base height and (b) the bias would be a function of cloud base height, meaning a characterization of the bias based on one evaluation dataset would only apply to datasets with the same distribution of cloud base heights. These are both clearly undesirable features, which we considered important enough to avoid that we accepted the added complication of nonlinear correction functions.

*#4: Page 6, lines 29-30: a) Do you confirm that you are introducing a new notation for RMSE, which is now "sigma"?*

Yes, for the reason hopefully now better stated in the manuscript: "The quality metric we use is the root mean square error (RMSE); the category RMSE determined from comparison to ceilometer $z$ then serves as the (sample) estimate of the predicted (population) standard deviation of the measurement error $z - \hat{z}$, i.e., the predicted column $z$ uncertainty. We denote this uncertainty as $\sigma_c$."

*b) Can you elaborate? For instance: what is the range of values for sigma(D,n,Dz)?*

*What is the sensitivity of sigma to D, n, and Dz?*

Due to the multivariate dependence on $D$, $n$, and $\Delta z$, this is hard to summarize succinctly. We have calculated one-dimensional projections of the full trivariate function onto each of the predictor variables, using the 2008 training dataset, similarly to what we have done in response to the question about the correction function. This additional figure is discussed in the revised manuscript.

In the R package that will be made public to accompany the final manuscript, the full predicted uncertainty table for each of the 125 categories in $D$, $n$, and $\Delta z$ is available.

*#5: Page 7, Equations 3 and 4: a) If I understand correctly, "z" in Eq. (3) is CBASE_z. Please clarify and use a specific notation for the different quantities. Indeed, "z" is used several times throughout the manuscript, but with different meanings.*

Good point. We now consistently use $z$ for the CBASE combined cloud base, $z_c$ for the local column cloud base from CALIOP, and $\hat{z}$ for the ceilometer cloud base.

*b) Please define "n" in Eq. (3).*

Done; thanks for pointing out the oversight.

*c) In Eq. (4), is sigma-i actually sigma-i(Di,ni,Dzi), with Di, ni, and Dzi associated to CALIOP local zi? How is ni defined for a local zi?*

Thank you for pointing out this source of confusion, which we have corrected. The revised manuscript notes that $D_i$ and $\Delta z_i$ are indeed properties of the individual profiles, but $n$ is a property of the group (and therefore carries no subscript).

*d) Can you discuss the impact of the training? For instance: how do CBASE_z and CALIOP mean base height compare for the year 2008 used to train the algorithm? How did you train the algorithm to have mean (CBASE_z -ceilometer $\hat{z}$) equal to zero (as suggested by statement line 23, page 7)?*

Unsurprisingly, the relationship between the CBASE cloud base and ceilometer cloud

base in the training dataset satisfies the constraints we designed into it: slope 1 and intercept 0 for linear regression, bias 0 and width 1 for $(z - \hat{z})/\sigma$. As this is more of a check that we do not have software bugs in the implementation, we did not consider it important enough to include in the manuscript, but we have added it now: "To check that the algorithm satisfies its design constraints, we have verified that linear regression between $z$ and $\hat{z}$ has zero intercept and unit slope, and that the quantity $(z - \hat{z})/\sigma$ has zero mean and unit standard deviation."

The real test is whether we overtrained the algorithm to fixate on peculiarities of the training dataset, which is the reason for testing on a statistically independent validation dataset, the results of which we discuss at length.

The removal of the bias is accomplished by be bias correction procedure; see our response to the question about the SVM-based correction method, where we have clarified the manuscript.

*#6: Page 8, lines 12-18: My understanding is that the algorithm training and the veri-fication presented in Sect. 3.4 using the 2007 data set have been carried out with no distinction between nighttime and daytime data. Did you investigate whether sigma-i (Di,ni,Dzi) are the same for nighttime and daytime data? I wonder whether the differ-ences between the nighttime and daytime CBASE_z highlighted here could be due in part to the fact that the algorithm training combines daytime and nighttime data.*

This is a very good point. While testing this hypothesis exceeds our resources at the moment, we have added discussion of this possibility to the manuscript: "Training a potential future update of the algorithm on daytime and nighttime profiles separately may reduce $\sigma$."

*#7: Page 8, lines 19-25: a) Figure 9 seems out of place. In my opinion, this discussion could be earlier in the manuscript. However, comparisons of 2B-GEOPROF-LIDAR and CBASE base altitudes would be informative.*

[Figure]

Agreed. We have added a subsection to the validation section that focuses on radar and lidar cloud base heights in 2B-GEOPROF-LIDAR compared to the same set of ceilometer measurements used to validate CBASE. We have also added a figure comparing the lidar-only 2B-GEOPROF-LIDAR and CBASE cloud bases explicitly. Unlike the CBASE cloud bases (which have been corrected), the 2B-GEOPROF-LIDAR lidar-only cloud bases are, on average, underestimates of the ceilometer bases for very low clouds ($<0.5$ km). The comparison of 2B-GEOPROF-LIDAR and CBASE similarly shows lower cloud base estimates by 2B-GEOPROF-LIDAR for very low clouds, and higher cloud base estimates for clouds above 1.5 km. The linear correlation between 2B-GEOPROF-LIDAR and CBASE is fairly good ($r = 0.79$).

*b) Please describe the "underlying physical measurement" in 2B-GEOPROF-LIDAR that explain the similarity of lidar-only 2B-GEOPROF-LIDAR and CBASE cloud bases.*

We simply meant that physically, both products are based on CALIOP attenuated backscatter, so the added value in CBASE comes from understanding the factors controlling profile-by-profile uncertainty. We have clarified the manuscript by explicitly stating this assumption.

*c) Are you implying that for lidar-only cases, cloud bases reported in 2B-GEOPROF-LIDAR differ from those reported in the V4.10 CALIOP VFM and are in better agreement with CBASE? Does 2B-GEOPROF- LIDAR use V4.10 CALIOP data?*

We did not intend to imply that; in fact, now that the reviewer has pointed it out, we realize we implicitly assumed that the 2B-GEOPROF-LIDAR and V4.10 CALIOP VFM cloud bases would be the same.

The version of 2B-GEOPROF-LIDAR we use (P2_R04_E02) uses VFM version 3 "or later", according to Mace and Zhang (2014), but the data files predate version 4, so presumably version 3.x was used. We have specified the 2B-GEOPROF-LIDAR version in the revised manuscript. Characterizing the differences between 2B-GEOPROF-LIDAR and the VFM would be outside the scope of this manuscript.

*#8: Page 9, lines 20-21: did the authors investigate how CBASE_z and CALIOP base altitude compare for clouds that are thick enough to attenuate the lidar laser beam?*

At the suggestion of the reviewer, we have included and discussed this figure in the revised manuscript (Figure 10); see the answer above to the question about 2B-GEOPROF-LIDAR and CBASE base altitudes would be informative.

*#9 Page 9, line 27: was stated but not shown.*

The statement in question is "The performance of CBASE $z$ is similar to that of 2B-GEOPROF-LIDAR lidar-only $z$ ...". We have clarified our intended meaning by adding "when validated against the same collocated ceilometer measurements". However, we agree with the reviewer's suggestion to include an explicit comparison of CBASE and lidar-only 2B-GEOPROF-LIDAR (see our answers above).

---

## Author Comment (AC2) · 9 Aug 2018

*In summary, I do not recommend this paper for publication in its current state. The methodology is questionable. The conclusions are highly qualified, and its not clear to me what exactly the authors have contributed to the science. In fact, its not clear to me at all why this methodology is even needed. Let me summarize my primary scientific concerns:*

The reason this methodology is needed is that all existing satellite cloud base products

have significant limitations. We do not claim that our product is flawless (and in fact a large fraction of the effort behind this product is dedicated to characterizing the errors); but science is an incremental endeavor, and the product incorporates enough features beyond existing products to make it a significant advance: cloud base heights, validated against ground observations, along the A-Train, for optically thick clouds, including validated point-by-point uncertainty estimates. We address the reviewer's specific concerns below.

*Its not clear why you'd need to resolve cloud base from CALIOP when CloudSat can do it. That's the whole point of the synergy between the lidar/radar.*

As we point out in the introduction, CloudSat is limited in its ability to detect cloud base because (a) the droplet size and thus radar reflectivity tends to decrease towards cloud base, frequently below the CloudSat detection limit, and (b) the lowest km of the profile tends to be affected by ground clutter. We have included references to these limitations of CloudSat in the introduction, and the manuscript includes a plot and a table documenting that CloudSat cloud base estimates perform worse than Calipso estimates even absent any attempts to correct or select high-quality lidar estimates. At the suggestion of Reviewer 1, the discussion of CloudSat cloud bases has been moved upward in the manuscript; we have also expanded upon the description of the CloudSat cloud base shortcomings in the introduction. We hope that this makes it clearer to the reader why we do not use CloudSat.

*Further, its not clear to me why, if you can resolve cloud base with CALIOP for an optically-tenuous clouds, why you'd need an algorithm to understand potential correlation and uncertainty, and thus who'd even use it?*

As we state at the beginning of the abstract (l. 2–4 of the manuscript), in the introduction, and again in the conclusions, we are not content to know the cloud base of optically tenuous clouds, but rather want to know the cloud base of optically thick clouds. We list several potential applications in the conclusions that indicate who potential users would be. As to why we would want to understand the uncertainty of a new product, we agree that not all new satellite products include a rigorous uncertainty analysis; however, we feel that this is important information to allow users to judge the quality of the product.

*If this paper is going to be publishable, the reviewers needs to go back and very considerately make the case that answers these questions. In my opinion, they have not done so beyond a threshold necessary for publication.*

In our opinion, the manuscript was already quite clear about why CloudSat and CALIOP without further processing are unsatisfactory for cloud base height. We have nevertheless tried to make the explanations even clearer in the revised manuscript.

*Training of your dataset relative to ground-based ceilometers, as you even state, limits your application to a very small set of cloud types. The authors seem aware of this, but only speculate as to its impact.*

It would be desirable to have a validation dataset for oceanic cloud in addition to continental, and we are very upfront about this in the manuscipt. That said, the range of cloud types observable over a year across the contiguous United States is not "very small". In our judgment, releasing a dataset with documented imperfections was preferable to polishing the apple forever. In particular, releasing the dataset makes it possible for others in the community to validate its performance for oceanic cloud if they are aware of a suitable validation dataset that we do not know of. (We note that such a validation exercise would be meaningful even if we did not retrain the algorithm on an oceanic dataset.)

*I ask again, who is the customer for this dataset, and how will it advance any scientific interest? How was the 100 km threshold for collocating with ceilometers chosen? What is the correlation length of cloud base spatially so as to justify such a choice? What is the impact on your results if you vary that threshold? There have been efforts (Omar et al. 2013 for aerosols...JGR-A) to collocate CALIOP with ground-based sun phot-*

[Figure]

*meters. They came up with something like 1500 suitable collocations under a much more stringent set of temporal and spatial thresholds. What you're trying to do requires far more justification and scientific basis, as it goes against conventional/proven thinking otherwise.*

The spatial decorrelation of the cloud base height is a good point. In principle, the algorithm learns in the training stage to give reduced weight to more distant measurements, as the RMSE increases with collocation distance, and increases the predicted uncertainty accordingly. In the revised manuscript, we have included a figure on the increase in predicted uncertainty as a function of distance (Figure 6). We find that that the lowest-uncertainty measurements do in fact come from the closest measurements ($D < 40$ km). Interestingly, 40 km is in fact the collocation threshold Omar et al. (2013) recommend (and the reason we have a larger number of collocations is that airport ceilometers vastly outnumber AeroNet stations).

We also note that we provide cloud base estimates with two collocation distance thresholds: 40 km and 100 km (stated in Section 4); the reason for this is to allow the user to make the tradeoff between increased probability that we can provide an estimate at a given location (the 100 km dataset; the smaller collocation threshold provides only approximately 1/5 as many collocations as the larger threshold) and lower cloud base uncertainty (the 40 km dataset).

*I recognize that as this is a Discussions page, that the likelihood is that the authors will be afforded opportunity to respond. That's fine. I caution, however, that if this were a more standard journal, I would be recommending an outright rejection.*

We are grateful for the opportunity to respond.